# Identification of Toxic Herbs Using Deep Learning with Focus on the Sinomenium Acutum, Aristolochiae Manshuriensis Caulis, Akebiae Caulis

**Jaeseong Cho [1], Suyeon Jeon [1], Siyoung Song [1], Seokyeong Kim [1], Dohyun Kim [1], Jongkil Jeong [1], Goya Choi [2] and Soongin Lee [1,*]**

[1] Department of Herbology, College of Oriental Medicine, Dongshin University, Naju 58245, Korea; tingstyle1@gmail.com (J.C.); suyeon.jeon@gmail.com (S.J.); yua0429@naver.com (S.S.); daisykim88@gmail.com (S.K.); ehgus9387@naver.com (D.K.); jgj3523@naver.com (J.J.)

[2] Herbal Medicine Resources Research Center, Korea Institute of Oriental Medicine, Naju 58245, Korea; serparas@kiom.re.kr

[*] Correspondence: barunhani@hanmail.net; Tel.: +82-10-6642-9571

**Abstract:** Toxic herbs are similar in appearance to those known to be safe, which can lead to medical accidents caused by identification errors. We aimed to study the deep learning models that can be used to distinguish the herb Aristolochiae Manshuriensis Caulis (AMC), which contains carcinogenic and nephrotoxic ingredients from Akebiae Caulis (AC) and Sinomenium acutum (SA). Five hundred images of each herb without backgrounds, captured with smartphones, and 100 images from the Internet were used as learning materials. The study employed the deep-learning models VGGNet16, ResNet50, and MobileNet for the identification. Two additional techniques were tried to enhance the accuracy of the models. One was extracting the edges from the images of the herbs using canny edge detection (CED) and the other was applying transfer learning (TL) to each model. In addition, the sensitivity and specificity of AMC, AC, and SA identification were assessed by experts with a Ph.D. degree in herbology, undergraduates and clinicians of oriental medicine, and the ability was compared with those of MobileNet-TL's. The identification accuracies of VGGNet16, ResNet50, and MobileNet were 93.9%, 92.2%, and 95.6%, respectively. After adopting the CED technique, the accuracy was 95.0% for VGGNet16, 63.9% for ResNet50, and 80.0% for MobileNet. After using TL without the CED technique, the accuracy was 97.8% for VGGNet16-TL, 98.9% for ResNet50-TL, and 99.4% for MobileNet-TL. Finally, MobileNet-TL showed the highest accuracy among three models. MobileNet-TL had higher identification accuracy than experts with a Ph.D. degree in herbology in Korea. The result identifying AMC, AC, and SA in MobileNet-TL has demonstrated a great capability to distinguish those three herbs beyond human identification accuracy. This study indicates that the deep-learning model can be used for herb identification.

**Keywords:** Herbology; herb identification; deep learning; Aristolochiae Manshuriensis Caulis; MobileNet; transfer learning

## 1. Introduction

As of 2015, there were 602 herbs mentioned in The Korean Herbal Pharmacopoeia, 618 herbs in the Pharmacopoeia of the People's Republic of China, and 215 herbs in the Japanese Pharmacopoeia [1]. Some toxic herbs are similar in appearance to other herbs, which is a matter of grave concern, as this may lead to drug-related accidents caused by mistake in visual discrimination. Therefore, the development of better herb identification technology is one of the most important and urgent tasks in herbal medicine.

The scientific names of Sinomenium acutum (SA), Aristolochiae Manshuriensis Caulis (AMC), and Akebiae Caulis (AC) are *Sinomenium acutum* (Thunb.) Rehder and E.H. Wilson, *Astolochia manshuriensis* Kom., and *Akebia quinata* (Houtt.) Decne. SA and AC have diuretic and anti-inflammatory effects, but AMC has a toxic medicine contains nephrotoxic and carcinogenic aristolochic acid [2]. The problem is that it is difficult to distinguish between them with the naked eye, as shown in Figure 1a. In fact, the accidental use of the toxic herb has resulted in drug-related mishaps in Korea, and the toxicity of AMC is the first to be mentioned in discussions on the toxicity of herbal medicines [3]. Since AMC is not distributed in Korea, practitioners do not have a chance to get to know AMC, so it is hard to notice that it has been replaced or mixed with AC or SA. This situation will be the same in other countries where AMC is not distributed. Therefore, for medical accidents caused by visual detection errors, deep-learning research can be a good alternative (Figure 1b).

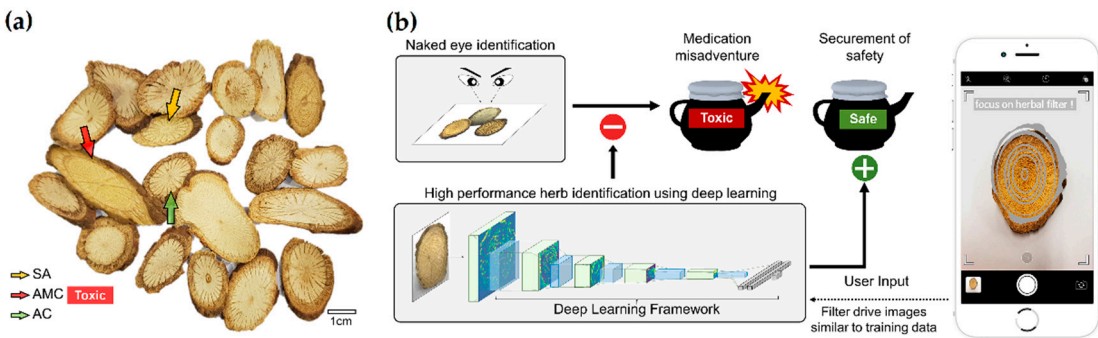

**Figure 1.** Necessity and search strategy for herb identification using deep learning (**a**) Sinomenium acutum (SA), Aristolochiae Manshuriensis Caulis (AMC), and Akebiae Caulis (AC) are difficult to distinguish with the naked eye; (**b**) we suppose that deep learning technology can reduce medical accidents caused by identification errors.

In Korea, it has been proposed that the characteristics of the tissue inside and outside the herbs can be distinguished using a microscope [4]. In China, a loop-mediated isothermal amplification method was proposed to distinguish between AC and AMC in 2016 [5]. However, it is more economic in terms of time and cost to distinguish herbs by appearance than by using a microscope or genetic analysis.

Recently, deep learning has been employed in computer vision, natural language processing, and reinforcement learning. These technologies have consequently replaced sensory evaluation in various fields. Research to identify medicinal herbs using modified VGGNet16 [6] and CafeNet [7] were conducted in the field of herbal medicine. To develop applications, it requires deep-learning technologies that are more accurate and operationally efficient. In particular, in order to be used as a mobile application in the future, the memory efficiency needs to be excellent. So, we found methods to differentiate AC, AMC, and SA with best accuracy using open source deep learning models, VGGNet16 [8], ResNet50 [9], and MobileNet.

## 2. Materials and Methods

### 2.1. Data Collection, Preprocessing and Augmentation

The herbs used on this paper are offered from Omniherb and Hanyaksarang Korean pharmaceutical companies. All raw herbs were judged to meet with the right scientific names by doctors with a Ph.D. degree in herbology. Images of herbs with white background were captured with a smartphone and were used in three deep-learning models to learn the specific characteristic of the herb. Additional images which were including background, blurred and overlapped were collected from the Internet. We collected real herb images in Google and Baidu, using Korean plant name, Chinese plant name, herbal name and scientific name as keywords. '청풍등', '青風藤' and 'Sinomenium acutum' were used for SA. '관목통', '關木通', 'Aristolochia Manshuriensis Caulis' and 'Astolochiae manshuriensis' were

used for AMC. '목통', '木通', 'Akebiae Caulis' and 'Akebia quinata' were used for AC. Accordingly, 500 images of the cut surfaces of each herb were captured with a smartphone. Each picture had a resolution of around 96–150 dpi. In addition, 100 images of each herb were obtained from the internet and cropped including background. Thus, 600 images were obtained for each of SA, AMC, and AC, resulting in a total of 1800 images. All images were resized into a square shape and flipping and rotation (−180° to +180°) were randomly performed to obtain a large number of images for the three herbs, as shown in Figure 2a. This constituted the data augmentation activity, which was the preprocessing step for training of the deep-learning models.

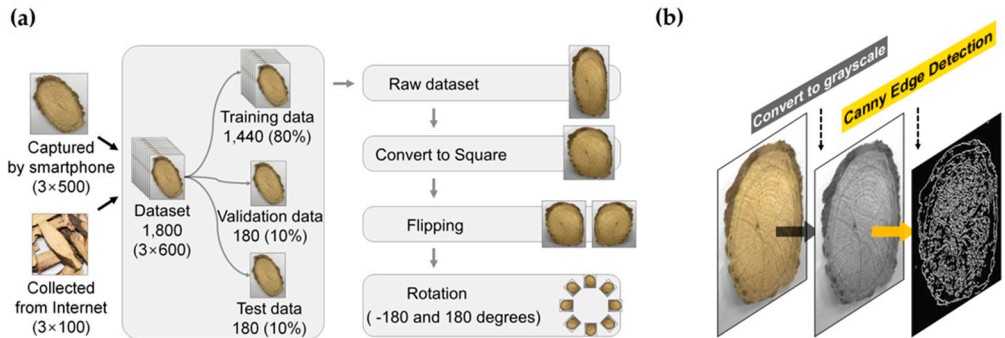

**Figure 2.** Strategy for developing high-performance herb identification deep-learning model. (**a**) Data preprocessing for deep learning; data augmentation used flipping and rotation; (**b**) process of canny edge detection (CED).

## 2.2. Deep-Learning Model Selection

We selected three convolutional neural network (CNN) learning models—VGGNet16, ResNet50, and MobileNet [10]—considering the possibility of implementing the proposed model as a mobile application (for completed models).

## 2.3. Learning and Validation

In order to evaluate the herb identification model, 1800 images were randomly divided into training, validation, and testing data in the ratio = 8:1:1. To prevent over-fitting during the learning process, an early stopping process [11] was employed based on the validation data. Furthermore, hyper-parameter was used without any tuning as in the study that initially introduced it.

## 2.4. Canny Edge Detection (CED)

To simplify the image data, all images of the herbs was converted to a grayscale image so that only the histological shape of the herb could be extracted from the edges [12] (Figure 2b). The performance of the VGGNet16, ResNet50, and MobileNet was compared before and after the application of canny edge detection (CED).

## 2.5. Transfer Learning (TL)

The transfer learning (TL) which is used in the ImageNet Large Scale Visual Recognition Competition (ILSVRC) assigns the weight of a deep learning model and adapts it to the data [13]. TL  was applied to VGGNet16, ResNet50, and MobileNet, and the performance in each case was compared with that before the application of TL.

## 2.6. Visualizing the Identification Weight of the Model

Gradient-weighted Class Activation Mapping (Grad-CAM) and Guided Grad-CAM were used to visualize the portion of the image that was weighted in the learning process [14].

*2.7. Indicators for Evaluating the Identification Models*

Accuracy is a standard metric for the classification performance of a deep learning model. Sensitivity and specificity, which are derived from the number of true positive (TP), false positive (FP), true negative (TN), and false negative (FN) results, are statistical measures of the performance of a binary classification test. TP, FP, TN, and FN are the components of a confusion matrix.

The receiver operating characteristic (ROC) curve is a graph with the true positive rate on one axis and the false positive rate on the other. This allows the performance of a deep-learning model to be assessed by means of lines on the graph and the human identification accuracy can be expressed as points calculated based on sensitivity and specificity. We used these indicators to compare the identification performance of the learning model with the human identification accuracy. These indicators are derived as follows:

$$Accuracy = \frac{TP + TN}{TP + FP + TN + FN} \tag{1}$$

$$Sensitivity = TP/(TP + FN) \tag{2}$$

$$Specificity = TN/(TN + FP) \tag{3}$$

*2.8. Survey Analysis for the Evaluation of Identification Abilities of the Learning Models*

An anonymous questionnaire was developed to evaluate the herbal medicine identification ability produced using Google questionnaire and the website link. The questionnaire was forwarded to 23 experts with a Ph.D. degree of the Korea Association of Herbology, 43 clinicians of the Jeonnam association of Korean Medicine, and 197 undergraduate students of the College of Korean Medicine, Dongshin University. On the first screen of the survey all participants were notified that they do not have to participate if they do not agree with the academic use of the survey data, which was used to compare the identification ability of artificial intelligence and humans. From 3 October to 8 October 2018, the number of people who voluntarily agreed participated to the survey was 129. Our survey was conducted on the basis of voluntary participation, was not coercive, and did not go through bioethical deliberations because they did not address sensitive personal information such as name, age, sex, phone number, address. Using the results of the survey, we analyzed the identification abilities of volunteers comprising six experts with a Ph.D. degree in herbology, 12 clinicians of Korean medicine, and 111 undergraduate students majoring in Korean medicine. The survey consisted of 60 questions, including 20 images of SA, AMC, and AC each. Prior information helpful in identification such as characteristic shapes of SA, AMC, and AC were presented before the questions. Then the images were presented to the volunteers and they were directed to select the appropriate image for each herb. The questionnaires had no time limitation, and redoing the questionnaires was not permitted. Each survey evaluated the sensitivity and specificity of each herb as described in the formula above (indicators for evaluating the identification models). These calculated sensitivity and specificity data were not normally distributed, so the Kruskal–Wallis test was conducted for comparison between groups of experts and visually compared with those of the best identification learning model based on the ROC curve.

*2.9. Training Environment*

We used a Windows 10 64-bit Intel core i7 processor with an NVIDIA GEFORCE GTX 1080Ti graphic card to train and evaluate the deep-learning models.

## 3. Results

### 3.1. Accuracy of the Three Convolutional Neural Network (CNN) Learning Models

The learning process involved the training of the VGGNet16 (31 epochs/12 min), ResNet50 (36 epochs/13 min), and MobileNet (52 epochs/17 min) using 1800 photos of SA, AMC, and AC. After the training process, 180 test images were entered to evaluate the identification accuracy of each model.

The average accuracy was 93.9% for VGGNet16, 92.2% for ResNet50 and 95.6% for MobileNet (Figure 3a). The accuracies were derived from confusion matrixes (Figure 3b). The confusion matrix shows what each model predicted as the proposed SA, AMC, and AC. The accuracy was calculated using the formula presented in the Materials and Methods section. The prediction accuracies of the herbs were 90.0%, 95.0%, and 93.3% for SA, 91.7%, 86.7%, and 95.0% for AMC, 100%, 95.0%, and 98.3% for AC, with VGGNet16, ResNet50, and MobileNet, respectively.

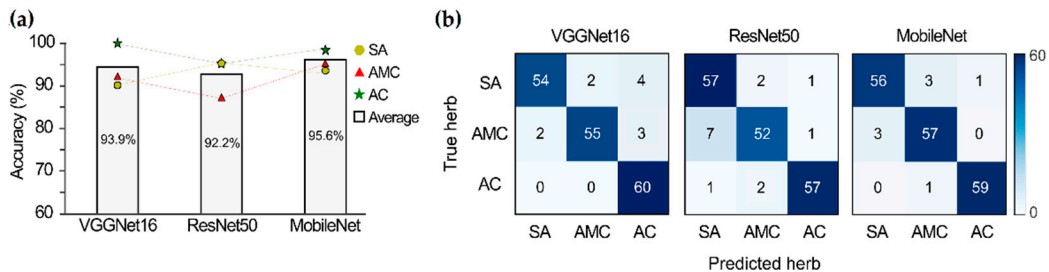

**Figure 3.** Performance evaluation of deep-learning model using raw images. (**a**) Identification accuracies for VGGNet16, ResNet50, and MobileNet; (**b**) Confusion matrix for each deep learning model. SA, Sinomenium acutum; AMC, Aristolochiae Manshuriensis Caulis; AC, Akebiae Caulis.

### 3.2. Evaluation of CED in Learning

The images were simplified by applying the CED algorithm (Figure 2b). The accuracy of VGGNet16 improved from 93.9% to 95.0%, but the accuracy of ResNet50 (92.2%→63.9%) and MobileNet (95.6%→80.0%) were lowered, as shown in Figure 4.

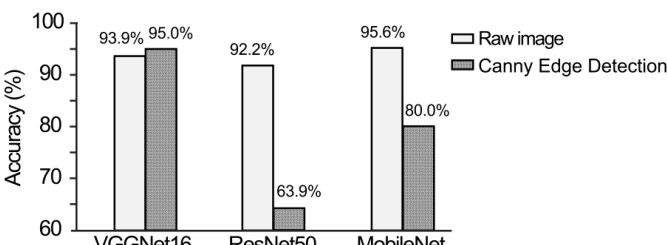

**Figure 4.** Performance evaluation of deep-learning model using edge extraction from images of herbal medicines. Comparison of the accuracies of each deep-learning model.

### 3.3. Visualization of the Weight of the Learning Model

We visualized some of the experimental data using the Grad-CAM and Guided Grad-CAM techniques, which visualized the predicted weights of MobileNet as a heat map. As a result, several cases were generated depending on the image being entered. In the first case (Figure 5a), weight was assigned to the histological shape of the cut surface, similar to the visualization by the human eye. In the second case (Figure 5b), weight was assigned only to the partial patterns of the cut surface. The third case (Figure 5c) focused on the entire image, including the appearance of the herb.

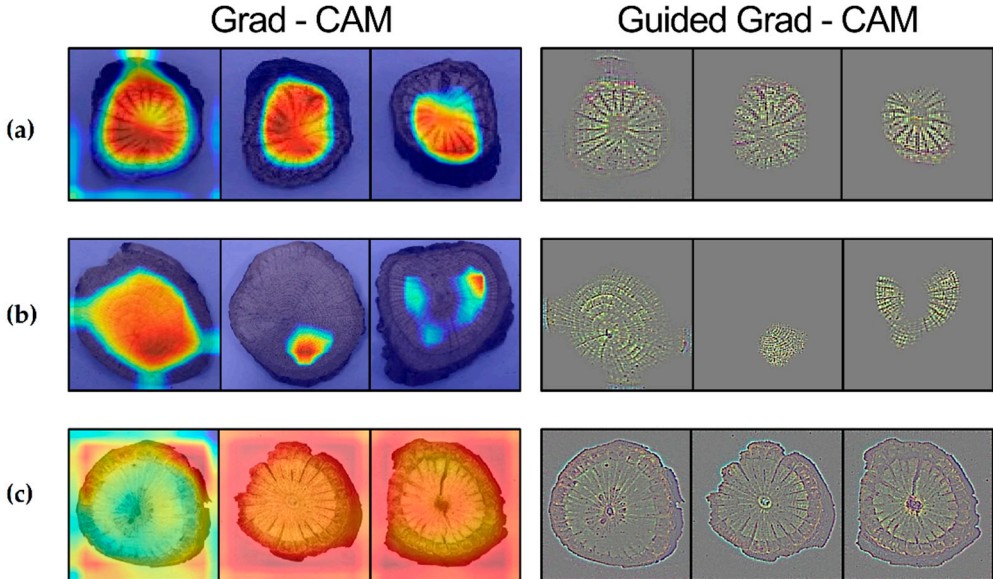

**Figure 5.** Visualization of learning weighted regions with MobileNet using Grad-CAM and Guided Grad-CAM. (**a**) Weight is assigned to the histological shape; (**b**) weight is assigned only to specific areas; (**c**) weight is assigned to the entire image including the appearance.

### 3.4. Usefulness of TL in CNN Learning

We applied TL to the deep-learning models to check whether it contributes to the accuracy improvement of the models. After training each model with TL, we evaluated the performance of each model.

The average accuracy of each model was 97.8% for VGGNet16-TL, 98.9% for ResNet50-TL, and 99.4% for MobileNet-TL (Figure 6a). This is a significant improvement over the accuracy of Figure 3a (93.9% for VGGNet16; 92.2% for ResNet50; 95.6% for MobileNet). The accuracies were derived from confusion matrices (Figure 6b). The confusion matrix shows what each model predicted for the proposed SA, AMC, and AC. The accuracy was calculated using the formula presented in the Materials and Methods section.

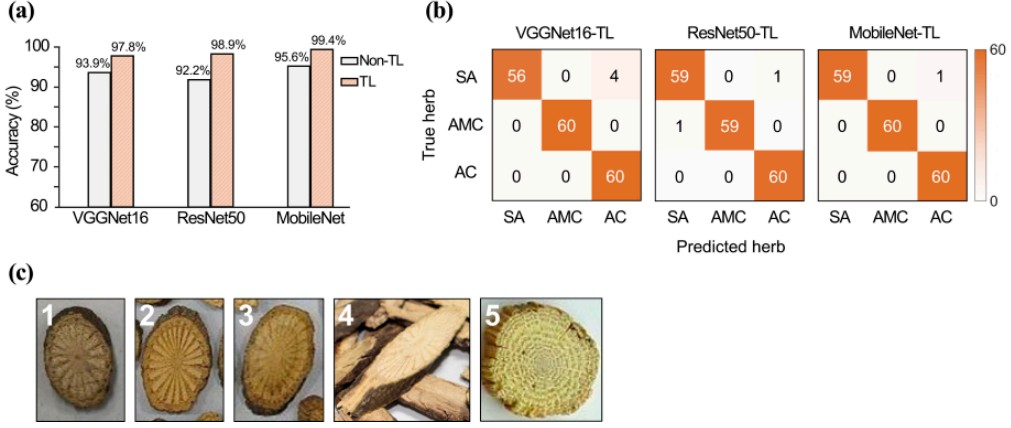

**Figure 6.** Performance of deep learning models trained by transfer learning (**a**) Accuracy of three deep learning models trained by TL; (**b**) Confusion Matrix of three deep learning models trained by TL; (**c**) Prediction failures: four images (Nos. 1–4) of SA predicted as AC; one image (No. 5) of AMC predicted as SA. TL, Transfer learning; SA, Sinomenium acutum; AMC, Aristolochiae Manshuriensis Caulis; AC, Akebiae Caulis.

The identification accuracy of the models improved, but prediction errors still occurred in five out of 180 images. The five images are shown in Figure 6c. VGGNet16-TL judged the four SA images Image Nos. 1–4 (Figure 6c, Image Nos. 1–4) as AMC. ResNet50-TL caused a prediction error on two images of an SA (Figure 6c, Image No. 4) and a peeled AMC (Figure 6c, Image No. 5). MobileNet caused only one error for the SA image (Figure 6c, Image No. 4), which led to a common failure for all three models. The SA image had blurred cuts, and displayed several herbs.

*3.5. Comparison of MobileNet-TL Discrimination Performance with Expert Discrimination Capability*

A survey was used to investigate the sensitivity and specificity of the identification of each herb by experts in the field, specially, volunteers with Ph.D. degree in herbology, clinicians of Korean medicine, and undergraduate students majoring in Korean Medicine. The identification results were visualized in comparison with that of MobileNet-TL, which showed the highest accuracy amongst all learning models in this study (Figure 6a). The red line in the figure is the ROC curve of MobileNet-TL; its area under the curve is 1.00 for all three herbs. The red, green and yellow data points denote the results for the Ph.D., clinicians and students, respectively. We chose 60 of 180 images used in the test for MobileNet-TL depending on their participation level in the survey, and used them in the survey for the identification performance of expert groups. Thus, this indicates that they were not analyzed or compared to each other under the same condition for the identification performance. Unfortunately, we could not examine statistical significance because we measured the accuracies of MobileNet-TL and experts under different conditions. However, the results for MobileNet-TL are likely to have greater ability in specificity and sensitivity than those of the expert groups. Figure 7 shows that the red circles, identification performance of Ph.D. for the three herbs, are few in number, yet are concentrated in the upper right.

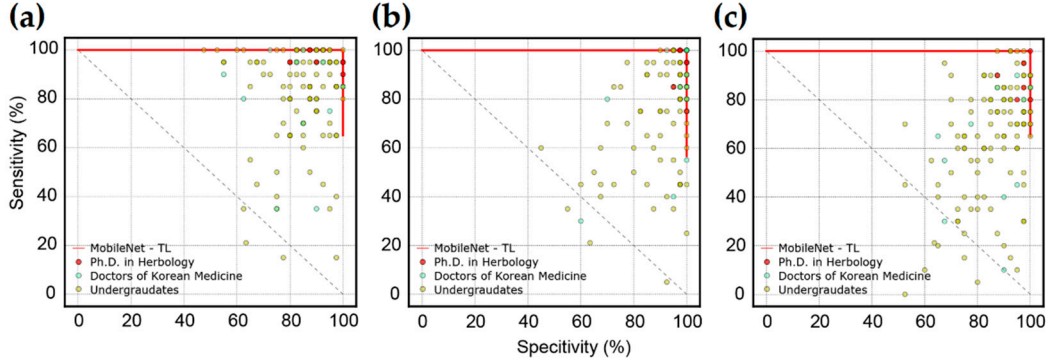

**Figure 7.** Comparison of the identification accuracies of MobileNet-TL and human expert group (**a**) Sensitivity/specificity for (**a**) SA; (**b**) AMC; (**c**) AC. Red line: receiver operating characteristic (ROC) curve of MobileNet-TL for 60 images of SA, AMC, and AC; circles: results for identifying 20 images of SA, AMC, and AC; Red circles: 6 experts with Ph.D. in herbology; green circles: 12 clinicians of Korean medicine; yellow circles: 111 undergraduate students majoring in Korean medicine. The darker the color, the more numerous. TL, transfer learning; SA, Sinomenium acutum; AMC, Aristolochiae Manshuriensis Caulis; AC, Akebiae Caulis.

Table 1 shows a comparison of sensitivity and specificity through the Kruskal–Willis test to check for significant differences in the points of each group on the ROC curve. It was found that sensitivity and specificity for SA, AMC, and AC are comparatively higher in Ph.D. than the ones in the other two groups. But there was no noticeable difference between the clinicians and the students. In consideration of sensitivity ($p = 0.023$) and specificity ($p = 0.053$) for AC, which is currently distributed in the market, there was a relatively significant difference among groups. Meanwhile, there was not much in AMC, which has been prohibited for distribution due to its toxicity.

**Table 1.** The sensitivity/specificity of the identification results conducted by the expert group.

| | SA | | | | AMC | | | | AC | | | |
|---|---|---|---|---|---|---|---|---|---|---|---|---|
| | Ph.D. (N = 6) | Clinicians (N = 12) | Students (N = 111) | $P$ | Ph.D. (N = 6) | Clinicians (N = 12) | Students (N = 111) | $P$ | Ph.D. (N = 6) | Clinicians (N = 12) | Students (N = 111) | $p$ |
| Sensitivity | 0.95 [0.95;0.95] | 0.88 [0.72;0.95] | 0.90 [0.75;0.95] | 0.310 | 0.95 [0.85;1.00] | 0.85 [0.68;0.95] | 0.85 [0.68;0.95] | 0.319 | 0.88 [0.80;0.95] | 0.68 [0.42;0.85] | 0.70 [0.50;0.80] | 0.023 * |
| Specificity | 0.95 [0.88;1.00] | 0.85 [0.74;0.91] | 0.85 [0.80;0.92] | 0.129 | 0.99 [0.97;1.00] | 0.99 [0.94;1.00] | 0.97 [0.90;1.00] | 0.509 | 0.97 [0.95;1.00] | 0.89 [0.73;0.95] | 0.88 [0.78;0.96] | 0.053 |

SA, Sinomenium acutum; AMC, Aristolochiae Manshuriensis Caulis; AC, Akebiae Caulis. Data are presented as median [1Q;3Q]. * $p < 0.05$.

## 4. Discussion

Deep learning is an artificial neural network consisting of several hidden layers between the input layer and the output layer. CNN is a type of artificial neural network that is commonly used to extract the representative features of an image [15]. In this study, the algorithms that solve the multi-class classification problem using CNN were applied to the identification of three types of herbs that are difficult to identify with the naked eye.

Learning models used for deep learning were developed in various ways. In this study, we selected the following three types of models: VGGNet16 (528 MB, TOP-5 accuracy: 71.3%) [8], ResNet50 (99 MB, TOP-5 accuracy: 74.9%) [9], and MobileNet (16 MB, TOP-5 accuracy: 89.5%) [10]. These models have a simple structure considering their performance and efficient memory use. VGGNet16 was proposed by Oxford University in 2014 and it offers better performance and memory efficiency by a single model than Google's GoogleNet proposed in the same year [8]. ResNet was proposed by an Microsoft researcher in 2015. Despite the fact that it has more layers than VGGNet16, it solves the poor learning performance of the inner layers, which is known as the degradation problem [9]. ResNet originally consisted of 152 layers. However, a ResNet50 model with 50 layers was used considering application distribution. MobileNet was proposed by Google in 2017. It has significantly reduced the amount of computation by using the depthwise convolution technique as opposed to the traditional convolution calculation method [10].

Prior studies [6,7] have attempted to identify herbs using deep learning in Table 2. We supposed that using refined image and applying the latest deep learning model is necessary to achieve better identification performance than that achieved with a group of experts.

As described in the Materials and Methods section, refined images and rough images of SA, AMC, and AC were collected. To check if there are false results in the data, data was randomly shuffled before the input. During the random image selection process, real herb images captured with smartphone and herb images collected from the Internet remained in a 5 to 1 proportion. The images of the herbs were augmented with flipping and rotation. VGGNet16, ResNet50, and MobileNet were learned with those images. The accuracies were as follows: VGGNet16: 93.9%, ResNet50: 92.2%, and MobileNet: 95.6% in Figure 3. Therefore, the three types of deep-learning models showed sufficient accuracy. It may be noted that we used close-up pictures obtained with the smartphones for deep learning, and this method of image acquisition appears to have contributed more strongly toward the learning of specific characteristics than the methods used in prior research. In general, various techniques for data augmentation can result in better learning performance by the model [16]. We used two skills of flipping and rotation for data augmentation, and they led to sufficient performances.

To ensure high accuracy, a strategy was employed to simplify the information in the images used for learning. This strategy is based on the fact that the appearance and histological shape of the cut surface of the herb are the key aspects examined in the identification performed by the naked eye. CED eliminates edges that are incorrectly detected by removing the non-maximum values that appear during edge detection, and finds sharper edges by using double thresholds [12]. The herb images we used are very sophisticated and have many textures, so we determined that the canny algorithm was more suitable than other edge detection algorithms such as Sobel and Laplacian.

Therefore, the accuracy was assessed by applying CED [12] to simplify the information, wherein only the histological shape was selected from the herb image. The results showed that the accuracy of VGGNet16 (93.9%→95.0%) did not vary widely, but those of ResNet50 (92.2%→63.9%) and MobileNet (95.6%→60.0%) showed a significant decline in Figure 4. There are several possible inferences for the decline in accuracy. First, not every deep learning model may perform well when a small number of images are used in the learning process, as the models employed in this study were proposed for the multi-classification capability of 1000 classes in ILSVRC. Second, the histological shape of the cut surface may not be a key element in the training process in deep learning. Third, image simplification can be an obstacle for deep-learning models with complex structures.

**Table 2.** Comparison of previous studies on herb identification using deep learning.

| Author/Year | Image | | | Training Model (Accuracy) | Comparison of Identification Ability with Expert Group |
| --- | --- | --- | --- | --- | --- |
| | Data Collection Method | Number of Identified Herbs (Number of Trained Images) | Background Inclusion | | |
| Sun and Qian/2016 | Internet images | 95 categories (5523 images) | Multiple herbs in an image with various backgrounds | Modified VGGNet16 (71.0%) | No comparison with experts |
| Weng et al./2017 | Images captured with smartphone | 11 categories (2200 images) | Single herb without any background | CafeNet (95.9%) | No comparison with experts |
| This Study | Images captured with smartphone and herb images from Google search engine | 3 categories (1800 images) | Most single herbs without backgrounds, and a few multiple herbs with backgrounds | VGGNet16-TL (97.8%) ResNet50-TL (98.9%) MobileNet-TL (99.4%) | The comparison with expert group conducted |

Grad-CAM and Guided CAM visualize the weights in the deep-learning model as a heat map, and reveal the data necessary to determine which portion of the image is weighted in image identification [14]. The learning weights for ResNet50 were analyzed using Grad-CAM and Guided CAM. In some pictures, weight was assigned to the histological shape of the cut surface, similar to the method followed in the identification with the naked eye (Figure 5a). In some other pictures, the weight was assigned only to a partial area of the cut surface (Figure 5b). In the rest, the focus was on the entire image, including the appearance of the herb (Figure 5c). This showed that the identification method in the deep-learning model was not the same as followed by humans and the same type of herb could be estimated by weighting in different patterns.

Another technique to improve the accuracy was to apply TL, which was presented in ILSVRC and is a learning technique that enhances the accuracy of the deep-learning model. It retrains the weight of a deep-learning model according to the data currently being learned. The accuracy of the three models prior to using TL were VGGNet16 93.9%, ResNet50 92.2%, and MobileNet 95.6%. By applying TL to the three deep-learning models, the accuracy improved to VGGNet16-TL 97.8%, ResNet50-TL 98.9%, and MobileNet-TL 99.4% (Figure 6a). The most accurate model was MobileNet-TL. Of the 180 images used in the test, one image had a prediction error in MobileNet-TL (Figure 6b). This image was low in quality and included several herbs, which were partially peeled, making it difficult to identify the herbs even with the naked eye (Figure 6c).

Finally, we compared the identification performance of MobileNet-TL with that of a group of experts, and its sensitivity for all three herbs was found to be higher for MobileNet-TL than for the group of experts (Figure 7). In addition, unlike our expectation that there will be differences in the identification ability among the three groups, we have not found significant differences other than the ones seen in AC. The explanation for this is that even experts have had little chance to encounter AMC since the distribution of the herb has been banned in 2005, and there has not been active and constant training for the clinicians in distinguishing such toxic herbs. Therefore, other means are needed to replace human sensitivity to AMC.

We present a deep-learning model that can be used for the toxic herb identification of AC, AMC and SA. We did not go through the optimization process by applying a wide variety of deep-learning methods or restructuring the deep-learning model. However, we suggest the possibility of reducing

the astronomical economic cost of simple discrimination by using the three open-source deep-learning methods to perform simple discrimination. The application of CED and TL to the three models of VGGNet16, ResNet50, and MobileNet resulted in the best performance being that of the MobileNet-TL, which learned images without CED. This study shows the possibility of the application of deep-learning techniques for the identification of various herbs.

**Author Contributions:** Conceptualization, S.L., J.C., J.J. and G.C.; Data curation, J.C.; Formal analysis, J.C.; Funding acquisition, J.J. and G.C.; Investigation, J.C., S.J., S.S., S.K. and D.K.; Methodology, S.L. and J.C.; Resources, S.L., J.J. and G.C.; Software, J.C.; Supervision, S.L., J.J. and G.C.; Writing—original draft, J.C., S.J., S.S., S.K. and D.K.; Writing—review and editing, S.L., J.C., S.K. and D.K.

**Funding:** This research was funded by the Herbal Medicine Resources Research Center, Korea Institute of Oriental Medicine, grant number K18401 and KSN1911420.

**Conflicts of Interest:** The authors declare no conflict of interest.

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
