# Peer review of "Identification of Toxic Herbs Using Deep Learning with Focus on the Sinomenium Acutum, Aristolochiae Manshuriensis Caulis, Akebiae Caulis"

_applsci, doi:10.3390/app9245456_

Round 1

Reviewer 1 Report

The manuscript is interesting, it presents the application of neural network and deep learning on herb recognition. The structure is clear, logic and easy to follow. I feel only minor adjustments are required, mainly in English, before acceptance.

Comments:

L25: "assessed for experts" -> "assessed by experts" L32: "And it was" please change the beginning of the sentence L33: "as" can be removed L41: "a matters" -> "a matter" L42: "accidents caused by visual discrimination." please clarify, probably authors mean mistake in visual recognition L55: "eye detection errors" -> "sensory/visual detection error" L64: "visual microscope or an optical microscope" unclear, please use only "microscope" or discuss with more details L66: "economical" -> "economic" L69: "humans" -> "sensory evaluation" L72-75: please give aim, not result at the end of introduction L79: "of" can be removed L80: "by the Ph.D. of herbology" please correct, PhD is a title only, it cannot do any activity L80: "without a background" please correct, all similar. Pictures have background, always. It might be "neutral background" or "white background". L87-88: can you provide the approximate resolution in mm/pixel? L98: please define CED but remove last part with CED and TL, since TL is not part of figure. L106: "study" maybe sample? L110: "black-and-white" please use grayscale like other places. BW means binary image. If so, please call it binary image. L117: "compared" -> "was compared" L131: "human ability to identify" please use another term, all similar. Maybe classification/recognition accuracy, etc. L144: "6, 12, and 111, respectively" unclear here, please use the sum since it is explained some lines later. L160-161: why graphic card is important? Did authors use GPU acceleration? L168: ", each" please clarify L190-191: "CED, Canny Edge Detection." can remove explanation, since it is not on figure. L206-207: "SA, Sinomenium acutum; AMC, Aristolochiae Manshuriensis Caulis; AC, Akebiae Caulis." can remove explanation, since it is not on figure. L260-264: please avoid repetition. L304: "enough performances." please use another term, maybe "sufficient ..." L336: "tried to compare" -> "compared"

Author Response

Thank you for your detailed comments.

Most of the points have been corrected according to your advice.

L25: "assessed for experts" -> "assessed by experts"

We corrected it.

L32: "And it was" please change the beginning of the sentence

We modified it to "And MobileNet-TL had higher identification accuracy "

L33: "as" can be removed

We corrected it.

L41: "a matters" -> "a matter"

We corrected it.

L42: "accidents caused by visual discrimination." please clarify, probably authors mean mistake in visual recognition

We corrected it.

L55: "eye detection errors" -> "sensory/visual detection error"

We corrected it to "visual detection error".

L64: "visual microscope or an optical microscope" unclear, please use only "microscope" or discuss with more details

We corrected it to "microscope".

L66: "economical" -> "economic"

We corrected it.

L69: "humans" -> "sensory evaluation"

We corrected it.

L72-75: please give aim, not result at the end of introduction

We described the definite aim of this study, as it follows “In particular, in order to be used as a mobile application in the future, the memory efficiency needs to be excellent. So, we found methods to differentiate AC, AMC, and SA with best accuracy using open source deep learning models, VGGNet16 [8], ResNet50 [9], and MobileNet.”

L79: "of" can be removed

We corrected it.

L80: "by the Ph.D. of herbology" please correct, PhD is a title only, it cannot do any activity.

We corrected it to "by doctors with a Ph.D. degree in Herbology ".

L80: "without a background" please correct, all similar. Pictures have background, always. It might be "neutral background" or "white background".

We corrected it to "white background ".

L87-88: can you provide the approximate resolution in mm/pixel?

We described it as “Each picture had a resolution of around 600 dpi.”

L98: please define CED but remove last part with CED and TL, since TL is not part of figure.

We modified the text of Figure 2(b), and cleared up the caption.

L106: "study" maybe sample?

We deleted “each study”.

L110: "black-and-white" please use grayscale like other places. BW means binary image. If so, please call it binary image.

We corrected it to "grayscale ". We did not mean “binary image”.

L117: "compared" -> "was compared"

We corrected it.

L131: "human ability to identify" please use another term, all similar. Maybe classification/recognition accuracy, etc.

We corrected it to “human identification accuracy”. We corrected L35, too.

L144: "6, 12, and 111, respectively" unclear here, please use the sum since it is explained some lines later.

We corrected it to "129 ".

L160-161: why graphic card is important? Did authors use GPU acceleration?

The specifications of the graphics card are important because they directly affect the reading time and identification performance. We did not use GPU acceleration.

Please refer to page 5/14 of the following paper.

- HAN, Seung Seog, et al. Deep neural networks show an equivalent and often superior performance to dermatologists in onychomycosis diagnosis: Automatic construction of onychomycosis datasets by region-based convolutional deep neural network. PloS one, 2018, 13.1: e0191493.

L168: ", each" please clarify

We corrected the sentence as “After the training process, 180 test images were entered to evaluate the identification accuracy for each model.”

L190-191: "CED, Canny Edge Detection." can remove explanation, since it is not on figure.

We deleted it.

L206-207: "SA, Sinomenium acutum; AMC, Aristolochiae Manshuriensis Caulis; AC, Akebiae Caulis." can remove explanation, since it is not on figure.

We deleted it.

L260-264: please avoid repetition.

We summed up the articles, as it follows “Comparison of the identification accuracies of MobileNet-TL and human expert group (a) Sensitivity/specificity for (a) SA; (b) AMC; (c) AC. Red line: ROC curve of MobileNet-TL for 60 images of SA, AMC, and AC; Circles: results for identifying 20 images of SA, AMC, and AC; Red circles: 6 experts with Ph.D. in Herbology; Green circles: 12 clinicians of Korean Medicine; Yellow circles: 111 undergraduate students majoring in Korean Medicine. The darker the color, the more numerous. TL, Transfer learning; SA, Sinomenium acutum; AMC, Aristolochiae Manshuriensis Caulis; AC, Akebiae Caulis.”

L304: "enough performances." please use another term, maybe "sufficient ..."

We corrected it.

L336: "tried to compare" -> "compared" 

We corrected it.

Reviewer 2 Report

The article presents a simple application of deep neural network techniques in the context of the recognition of toxic herbs.

Three different well known networks were compared on images of herbs segmented from the background. Further tests were done with an edge extraction preprocessing, but which led to a decrease in classifier performance. Finally, the application of transfer learning has shown an increase in results.
In my opinion the results obtained are obvious for the scientific community. The deep networks have amply demonstrated that they can work directly on images without preprocesing, as well as  the TL is known to produce improvements in classification.

The experimental results section lacks critical comments  as well as the comparisons with classification results obatined by human experts is not significant. These people are not really expert in classification tasks (undergraduate students, Ph.Ds, and some clinicians) as it not clear if they have had experience in identification of these toxic herbs. 

Author Response

Your accurate point has given us the opportunity to reflect on our research methods. It was a good opportunity to make up for the part that was not explained in the paper submitted.

1. In my opinion the results obtained are obvious for the scientific community. The deep networks have amply demonstrated that they can work directly on images without preprocessing, as well as the TL is known to produce improvements in classification.

We agree to your opinion. Our research methodology is not special at all in the field of deep learning because it uses an easy-to-use and obvious deep learning technique. However, these three herbs are not easy to discriminate with the naked eye. We’d like you to consider it an important issue in the field of Korean medicine that treats life of a person. Simple mistake or intentional fraud on discriminating poisonous herbs leads to serious medical accidents. Currently, a myriad of economic costs are used only for the simple differentiation of herbs in the Korean medicine community. We are trying to develop practical applications for thousands and hundreds of herbs. We have determined that it is necessary to inform the Korean medical community that the discrimination of high difficulty can be achieved through the “unspecial” technique.

As you point out, we have described a critical comment in our discussion that the learning techniques used in this study are not special at all, and that the optimizations obtained from comparisons with other numerous techniques were not sufficiently achieved.

2. The experimental results section lacks critical comments, as well as the comparisons with classification results obatined by human experts is not significant. These people are not really expert in classification tasks (undergraduate students, Ph.Ds, and some clinicians) as it not clear if they have had experience in identification of these toxic herbs.

Reflecting your point of view, we added a comment stating that we were unable to statistically examine superiority of human discrimination ability and MobileNet-TL, for the conditions for measuring MobileNet-TL and expert's identification accuracies were different.

Since AMC is not even distributed in Korea, the ability of experts to discriminate AMC “in the long term" will be reduced in the future. The three groups of experts used here are those who have or are receiving major education at universities that include identification practices for SA, AMC, and AC. In other words, as those who have a clear understanding of toxic herbal medicines, the Republic of Korea government narrowly grants them the right to distribute or clinically use herbal medicines. Please consider these facts.

Reviewer 3 Report

How was erroneous results resulting from a false local minimum secured? Have you tried to use the Monte Carlo method to select test images for this purpose?

Why from the whole group of edge detection algorithms only Canny was used. Maybe others would be better? A comparison with other edge detection algorithms is required.

Table 2 is difficult to read.

Author Response

Your accurate point has given us the opportunity to reflect on our research methods. It was a good opportunity to make up for the part that was not explained in the paper submitted

1. How was erroneous results resulting from a false local minimum secured? Have you tried to use the Monte Carlo method to select test images for this purpose?

Since we cannot check the false results if we do shuffling at the input stage, we randomly shuffled the data and secured the label before input. In the process of randomly extracting 180 test images, the Monte Carlo method was not considered in order to maintain a 5:1 ratio (which is described in material and methods part) between the photographed images and the real herb images collected from the Internet. The attached images below are the wrong images of the VGG16-TL which were collected from secured label.

Figure 1. the wrong images of the VGG16-TL (Please see the attatchment)

We also found the optimal optimizer for each deep learning model to avoid a false local minimum, and we used the LearningRateScheduler, a callback function provided by keras, to find the optimal loss value. EalryStopping in the section where loss value continuously decreases prevented the model from overfitting the data. After those efforts, we could find the learning curves in MobileNet-TL and ResNet50-TL (as you can see in Figure 2 below), which performed best. Through those strategies, we avoided a false local minimum.

Figure 2. Learning curves of MobileNet-TL (a) and ResNet50-TL (b). (Please see the attatchment)

2. Why from the whole group of edge detection algorithms only Canny was used. Maybe others would be better? A comparison with other edge detection algorithms is required.

In our study, we used python language (ver 3.6) and openCVPakage. And OpenCV is the most widely used Pakage for image data preprocessing of Deep learning. This Pakage provides Edge detection algorithms including Canny, Sobel, and Lapalcian etc. Among these algorithms, Canny Edge Detection removes edges that are incorrectly detected by removing the Non-Maximum value that appears during edge detection, and can find sharper edges by using double threshold. Since herbs images that we used are very sophisticated and have a lot of textures, we decided that the best algorithm for edge detection for this study would be the Canny algorithm. This is described in reference (12. Canny, J. A computational approach to edge detection. IEEE Trans. Pattern Anal. Mach. Intell. 1986, 6, 679–698.) As shown in the figure below (Figure 2), the Sobel or Laplacian algorithms use only the magnitude of the gradient vector, so the edges are detected thicker compared to Canny algorithm. In the end, we also attempted learning using the edge detection with CED, but the results suggested that the accuracy was lowered. And this attempt is summarized and reflected in the discussion.

Figure 3. Edge detection by Canny, Sobel, Laplacian. (Please see the attatchment)

3. Table 2 is difficult to read.

To improve the readability of Table 2, we changed the alignment in the table to left alignment and added a horizontal line.

Round 2

Reviewer 3 Report

Thank you for developing comprehensive answers to your questions. I would recommend, in the future, closer observation of the impact of detection on the results of your research.